# A new high-resolution groundwater isoscape for South-East Germany: insights from differences to precipitation

Aixala Gaillard[1], Robert van Geldern[1], Johannes Arthur Christian Barth[1], Christine Stumpp[2]

[1] Friedrich-Alexander-Universität Erlangen-Nürnberg, Department of Geography and Geosciences, Geozentrum Nordbayern, Schlossgarten 5, 91054 Erlangen, Germany
[2] BOKU University, Institute of Soil Physics and Rural Water Management, Department of Landscape, Water and Infrastructure, 1190 Vienna, Austria
*Correspondence to*: Aixala Gaillard (aixala.gaillard@fau.de)

**Abstract.** Stable water isotopes are important tracers to understand interactions between all compartments of the hydrological cycle. Particularly in groundwater studies they have been used to assess recharge and origin of groundwater. Based on high-density groundwater measurements with 596 measurements stations in an area of 70,500 km$^2$, we produced a new, interpolated map of the $\delta^{18}O$ distribution (i.e. an isoscape) in groundwater of south-east Germany. A comparison of this groundwater isoscape to the regional long-term precipitation isoscape showed differences of up to ±2 ‰ between both compartments. Groundwater was enriched in the light isotope $^{16}O$ compared to precipitation in the alpine area where recharge occurs mainly during winter and at recharge areas at higher altitudes. However, groundwater was enriched in the heavy isotope $^{18}O$ with respect to precipitation in drier lowland regions where summer recharge, irrigation practices and aquifer types as well as evaporation processes may play a role. Further comparative studies between precipitation and groundwater stable water isotopes including time series are needed to further improve the understanding of spatial and seasonal recharge patterns of groundwater.

**Graphical abstract.**

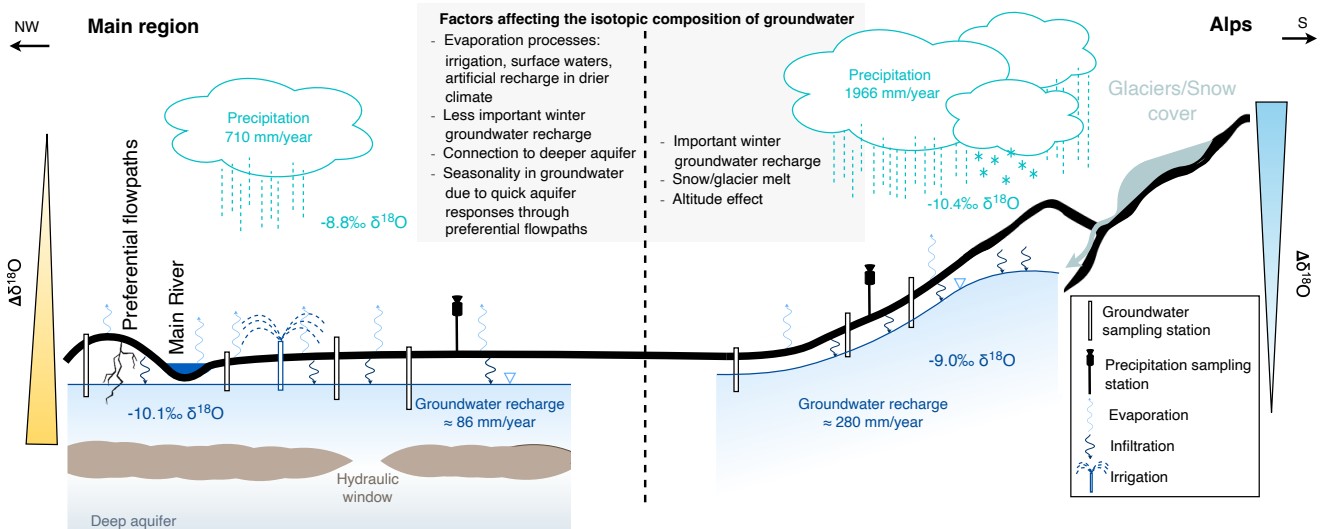

## 1 Introduction

Stable isotope ratios of oxygen ($^{18}O/^{16}O$) and hydrogen ($^{2}H/^{1}H$) in the water molecule have long been used as tracers in the
hydrological cycle (Rozanski et al., 1993; Dansgaard, 1964; Craig, 1961). For groundwater studies, they can provide new
information about the origin of precipitation or about surface water-groundwater interactions (e.g., Koeniger et al., 2016). Such
insights contribute to improve knowledge about groundwater recharge, both in terms of spatial patterns and seasonality (Bowen
et al., 2019). Other benefits include better quantification of infiltration, evaporation, and groundwater vulnerability to surface
pollutants (Ju et al., 2024; Lorenzi et al., 2024).

For a long time, only localized measurements were available for groundwater isotope studies, although continuous spatial
information is needed to reach an overall understanding of flow directions and recharge processes (Bowen, 2010). Maps
representing the spatial variation of isotopes, so-called "*isoscapes*", have been developed and improved since the 2000s with
the help of Geographic Information Systems (GIS). Such developments have been made possible with the increased availability
of extensive isotope datasets, mainly in precipitation, and high-resolution climate data. For instance, similar patterns strongly
connect isotopes in precipitation to those in the biosphere (Hobson et al., 1999; Bowen et al., 2005; Bowen and Revenaugh,
2003; Bowen and Wilkinson, 2002; West et al., 2008a; Wassenaar and Hobson, 1998; Bowen, 2010). Such isotope relationships
offer help in various fields that range from forensics via archaeology, the food and beverage industry, to animal migration
investigations (West et al., 2008b; Cerling et al., 2016), but also to understand hydrological processes.

A global precipitation isoscape was established with the data from the Global Network of Isotopes in Precipitation (GNIP)
(Terzer et al., 2013; Terzer et al., 2021). In the interpolation model, effects on isotope ratio variability such as latitude, altitude
and climate including precipitation quantity, vapor pressure and longwave radiation are accounted for. Parallel to this global
approach, several other local and regional interpolations of isotopes in precipitation have been proposed (Lykoudis and
Argiriou, 2007; Delavau et al., 2015; Hollins et al., 2018).

Much fewer isoscape investigations exist for groundwater. So far, some regional groundwater isoscapes have been presented
for Mexico (Wassenaar et al., 2009), the Baltic region of Estonia, Latvia and Lithuania (Raidla et al., 2016), Ireland (Regan et
al., 2017), Poland (Leśniak and Wilamowski, 2019), Croatia (Brkić et al., 2020) and Ethiopia (Bedaso and Wu, 2021).
However, comparing precipitation and groundwater isoscapes may reveal groundwater recharge areas and their complex
recharge mechanisms that include post depositional evaporation or recharge from surface waters. Such comparisons can also
provide services for interpretations of seasonal or long-term temporal trends in the context of climate change if repeated over
time. For example, groundwater isotope values of paleowaters have been used in combination with groundwater mean residence
times to reconstruct paleo-recharge conditions and quaternary climate change (Jiráková et al., 2011; Darling, 2011; Edmunds,
2001; Négrel and Petelet-Giraud, 2011). So far, such comparisons have only rarely been applied at a larger, regional scale.
Most aforementioned groundwater isoscapes underline the need for higher data density and an unbiased and even distributions
of datapoints across the study area and aquifers as well as consistent sampling and analysis methods. This is especially relevant

to ensure valid interpolation models across larger investigation areas that take into account regional climatic and geologic particularities (Bedaso and Wu, 2021).

Generally, we expect the isotopes in groundwater to represent a long-term average of the isotopes in precipitation as seasonal patterns are filtered out during infiltration and mixing with the large, long-term reservoir of groundwater. This implies that groundwater isotope patterns can serve as proxies for precipitation (Wassenaar et al., 2009; Raidla et al., 2016; Bedaso and

Wu, 2021). Nonetheless, groundwaters in temperate regions can show a bias towards the lighter isotopes of winter precipitation. This is due to well documented groundwater recharge during winter, mostly because of smaller evapotranspiration during the non-growing season (Riedel and Weber, 2020; Clark, 1997; Négrel and Petelet-Giraud, 2011). Studies in Poland, Germany and France have estimated that 0–51 % of winter precipitation, but only 0–17 % of summer precipitation turn into groundwater (Jasechko et al., 2014; Raidla et al., 2016).

The opposite situation, when groundwater displays higher isotope composition than precipitation can be explained by several mechanisms. First, evaporation processes can enrich the remaining groundwater in heavy isotopes after precipitation reached the ground. This can happen during water infiltration into the ground, at surface waters (lakes, rivers and artificial recharge ponds) that themselves feed the groundwater, or when large amounts of groundwater are brought back to the surface, for example for irrigation purposes (e.g. Yousif et al., 2016). Similarly, infiltration of tap and/or wastewater with a different

isotopic signal through leakage in the canalisations may also contribute to more positive values in the groundwater. Second, a bias in groundwater towards summer precipitation is possible if recharge takes mostly place during the warm season. This is possible in regions where the potential evapotranspiration is roughly equal to the precipitation, i.e. in arid areas. In this case, isotopically enriched high intensity rain events taking place during the warm season disproportionately contribute to the distinct isotope signal observed in groundwater due to preferential flowpaths (Wheater et al., 2010). Moreover, a connection of the

shallow aquifer to deeper groundwater may change isotope ratios in comparison to precipitation. Finally, isotopes in groundwater may also show dampened seasonal variations at a local scale in aquifers with short transit times and low dispersivities (Raidla et al., 2016).

The non-equilibrium process of evaporation is described quantitatively by the Craig-Gordon model (Craig and Gordon, 1965). According to the C-G model, evaporation effects can typically be seen in the dual-isotope plot ($\delta^{18}O$ vs $\delta^2H$) by sample isotope

ratios that plot along a lower slope with respect to the Meteoric Water Line (MWL).

A secondary parameter derived from the linear relation of isotope ratios in global precipitation, known as the Global Meteoric Water Line (GWML) (Craig, 1961) is the deuterium excess (*d*) value, introduced by Dansgaard (1964) with:

$$d = \delta^2H - 8\,\delta^{18}O \qquad\qquad\qquad (1)$$

This parameter is known to vary locally due to kinetic fractionation processes sensitive to relative humidity and advection by winds at the moisture source (Clark, 1997). Globally, *d* values range around 10 ‰ worldwide (Craig, 1961). In German

precipitation, weighted annual means for *d* have been reported between 4.1 and 11.9 ‰ (Stumpp et al., 2014).

With this work we investigated similarities and differences between groundwater and precipitation stable water isoscapes in a high-resolution study in southern Germany. The objectives of this study were: (i) to create a high-resolution regional isoscape for shallow groundwater in the south-eastern part of Germany (Bavaria), (ii) to compare this groundwater isoscape to precipitation isotopes in the same area, and (iii) to identify the mechanisms that explain the possible local or regional differences between isotopes in precipitation input and groundwater. The investigated area represents a typical central European temperate climate with respect to land use, topography, altitudinal gradient and urbanization. The groundwater data set offers an exceptionally high spatial resolution and equally covers the main shallow aquifers. Thus, this study offers new opportunities to investigate two crucial components of the hydrological cycle, that are precipitation and groundwater on a regional level.

## 2 Methods

### 2.1 Study area

Our study focuses on the south German state of Bavaria in central Europe (Fig. 1a). It covers 70,500 km$^2$ from the Alps and alpine foreland in the south (maximum altitude 2,962 m above sea level; a.s.l.), parts of the Danube Basin and the low-lying Main River region in the northwest (about 100 m a.s.l.). The main aquifers in the Alps occur in valley sediments. North of the Alps, the Molasse Basin is characterized by aquifers with flow occurring though either fractures or pores. Karst systems are characteristic for central Bavaria. These are bordered to the east by low permeability crystalline rocks. The region of the Main River in the northwest mainly presents fracture aquifers in sand- and limestone formations (Fig. 1c, Bundesministerium für Umwelt Naturschutz und Reaktorsicherheit, 2003).

The study area lies in a temperate continental climate with the code Dfb according to the Köppen-climate zones (Beck et al., 2020). Average annual precipitation and temperatures are 941 mm and 9.7 °C (Bayerisches Staatsministerium für Umwelt und Verbraucherschutz, 2021) with strong regional variations. For instance, the Main River region showed an annual average precipitation of 710 mm in the period 1971-2000. This strongly contrasts with the Alps with annual precipitation average of 1966 mm for the same period. Similarly, the groundwater recharge for the years 2009 to 2018 amounts to 280 mm/year in Rosenheim and 86 mm/year in Aschaffenburg. These measurements are representative for the alpine and Main River regions, respectively (Bayerisches Landesamt für Umwelt, 2018, 2020).

### 2.2 Groundwater sampling and analysis

Groundwater sampling took place from April to August 2015 in collaboration with the local water management authorities of Bavaria. A total of 596 water samples were collected in wells ($n$=402) and springs ($n$=194) over the entire study area (Fig. 1a) according to standard groundwater sampling protocols. While few deeper wells were also sampled, mostly shallow aquifers with depth < 40 m below surface were addressed.

All water samples were analysed for stable isotope ratios ($^{18}O/^{16}O$ and $^{2}H/^{1}H$) with an isotope ratio infrared spectroscopy

analyzer (Picarro Inc. Santa Clara, CA, USA). Results are denoted in the standard delta-notation for the relative difference of

isotope ratios in permille (‰):

$$\delta^i E = \frac{R(^iE/^jE)_{sample}}{R(^iE/^jE)_{reference}} - 1 \tag{2}$$

Where $R$ represents the dimensionless ratio of heavy isotopes (mass $i$) to light isotope (mass $j$) on an element $E$ for a sample

or a reference substance (Coplen, 2011). Raw data values were normalized to the Vienna Standard Mean Ocean Water

(VSMOW) scale and corrected for sample-to-sample memory as well as instrument drift according to van Geldern and Barth

(2012). The long-term measurement precision represented as standard deviation ($\pm 1\,\sigma$) on independent control samples was

< 0.1 ‰ and < 1.0 ‰ for $\delta^{18}O$ and $\delta^{2}H$.

### 2.3 GNIP data acquisition

The monthly water isotope values in precipitation from 2010 to 2018 were retrieved from GNIP stations in and around Bavaria

(accessible from the WISER database, https://nucleus.iaea.org/wiser (IAEA, 2023b) and measured by ourselves (Stumpp et al.,

2014). The gridded interpolation maps of $\delta^{18}O$ values averaged over an entire year were downloaded from the IAEA-Isotope

Hydrology Network (IHN) website in September 2023 (https://isotopehydrologynetwork.iaea.org (IAEA, 2023a). The

interpolation model for these isoscapes is described in Terzer et al. (2021).

### 2.4 GIS and statistical methods

All maps were produced with ArcGIS Pro 2.9.0 using the Germany Zone 4 projected coordinate reference system

(ESRI:31494). Based on the assumption that altitude and temperature differences are negligible in groundwater, that latitude

and the distance to oceans are irrelevant on this spatial scale and that our samples were evenly distributed over the entire area,

we opted for a purely geostatistical approach to produce a groundwater isoscape. We used the "Empirical Bayesian Kriging"

algorithm from the Geostatistical Analyst Toolbox. Input parameters for the algorithm can be found in the supplementary

information (Table S1). This algorithm proposes an improvement of traditional Kriging by estimating the uncertainty of the

semivariogram model and automating parameters best fit (more information in the ArcGIS Pro documentation

(https://pro.arcgis.com/en/pro-app/2.9/tool-reference/geostatistical-analyst/empirical-bayesian-kriging.htm       (ArcGIS Pro,

2023). Compared to traditional Kriging, this improved algorithm proved less sensible to outliers and reduced the number of

local artefacts around outlier sampling points. The standard error of the produced surface results from the standard error of the

interpolated values. Data transformation and statistics were performed with R (version 4.3.0).

### 2.5 Cluster attribution

To display differences between  isotope values in groundwater and precipitation, we subtracted the precipitation isoscape from

the groundwater isoscape for both $\delta^{18}O$ and $\delta^{2}H$ by using the arithmetic raster function built in ArcGIS Pro. Due to the analysis

error ($\pm$ 0.1 ‰ for $\delta^{18}$O-values) and the uncertainty of the interpolations ($\pm$ 0.3 ‰ for groundwater, see Sect. 3.1, $\pm$ 0.1 ‰ for

precipitation), we considered groundwater/precipitation differences under $|0.5|$ ‰ $\delta^{18}$O ($|5.0|$ ‰ $\delta^2$H) as not significant and therefore did not further interpret them. The areas where the difference between groundwater and precipitation exceeded this threshold were grouped into two clusters: a) one in which the isotope values in groundwater were more negative than in the precipitation (blue cluster) and b) one for the opposite situation (yellow cluster). We then attributed the groundwater sampling points to a cluster when the point lay within 25 km to this cluster. To compare to precipitation data, we also considered 13

GNIP stations with at least 8-year monthly time series in and around Bavaria. These were attributed to the clusters based on the same geographic criteria as for groundwater. The location of the considered GNIP stations and their cluster attribution can be found in Fig. 2.

## 3 Results

Here, we present isoscapes of the $\delta^{18}$O values distribution in groundwater and precipitation. Distribution maps of the hydrogen

isotopes proved similar and are displayed in the supplementary information (Figure S1).

### 3.1 Groundwater and precipitation data

Figure 1b presents the $^{18}$O-isoscape of Bavarian shallow groundwaters. The groundwater $\delta^{18}$O values ranged from -12.9 to -6.8 ‰ with a mean value of -9.6 ‰. We observe a clear trend of more negative isotope values in the south (alpine region) to less negative values in the north-western Main region. The standard deviation associated to the laboratory analysis of $\delta^{18}$O of

< 0.1 ‰ lies at least one order of magnitude below the observed spatial trends of several permilles. The standard error of the interpolation ranged from 0.1 ‰ to 0.6 ‰, which is comparable the analytical error. However, this uncertainty lies under 0.3 ‰ in >95 % of our study area (Fig. 1d).

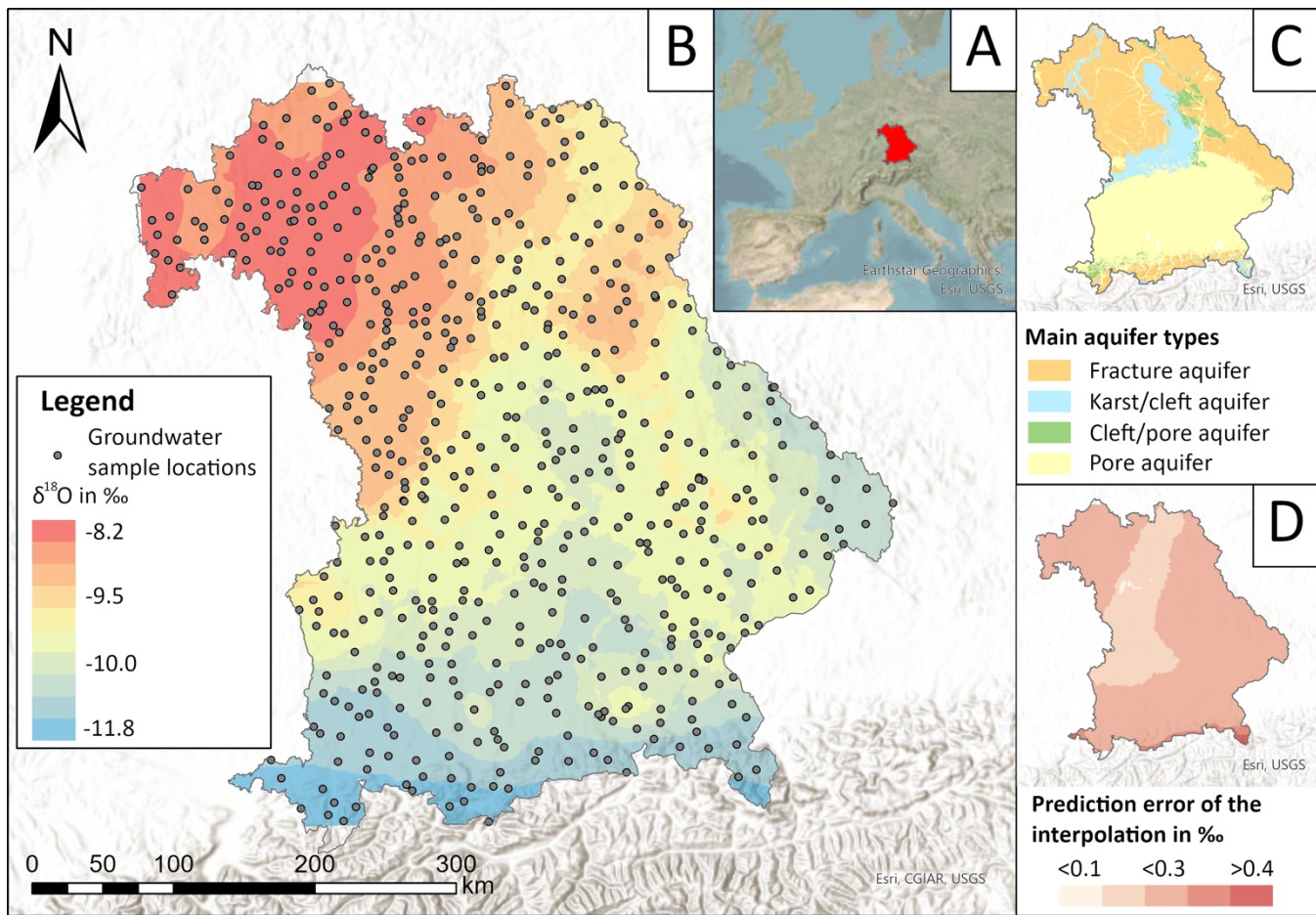

Figure 1: (a) Location of the study area in Europe. (b) Groundwater $\delta^{18}O$ isoscape (obtained with empirical Bayesian Kriging) with sample locations. (c) Main hydrogeological units in Bavaria according to the Hydrologeological Map for Germany (HÜK250) provided by the German Federal Institute for Geosciences and Natural Resources (Bundesanstalt für Geowissenschaften und Rohstoffe and Staatlichen Geologischen Dienste, 2019). (d) Standard error of the interpolation prediction in ‰.

The distribution of $\delta^{18}O$ values in precipitation as obtained by the interpolation by Terzer et al. (2021) is shown in the supplementary information (Figure S2). The data range from -13.4 to -8.8 ‰. The distribution pattern in precipitation depends on the altitude with the lowest values in the Alps and the eastern regions bordering the study area.

### 3.3 Groundwater–precipitation comparison

Figure 2 displays the difference of $\delta^{18}O$ values between groundwater and precipitation. About 67 % of our investigation area, mainly central Bavaria, showed less than ±0.5 ‰ difference between groundwater and precipitation. When this was the case, we concluded that precipitation recharges groundwater equally throughout the year with little or no alteration of the isotope

signal and did not further consider these regions in the subsequent discussion that focuses on isotope differences ($\Delta\delta$-values). Areas with substantial differences between isotopes in precipitation and groundwater include the Alps, alpine foreland and molasse basin (19.3 % of the study area) with groundwaters having more negative values than the local precipitation, with a maximum difference of 2.1 ‰. In contrast, the north-western part of Bavaria that roughly corresponds to the Main region (13.3 % of the study area) showed more negative values in precipitation than in groundwater, with a maximum difference of 1.9 ‰.

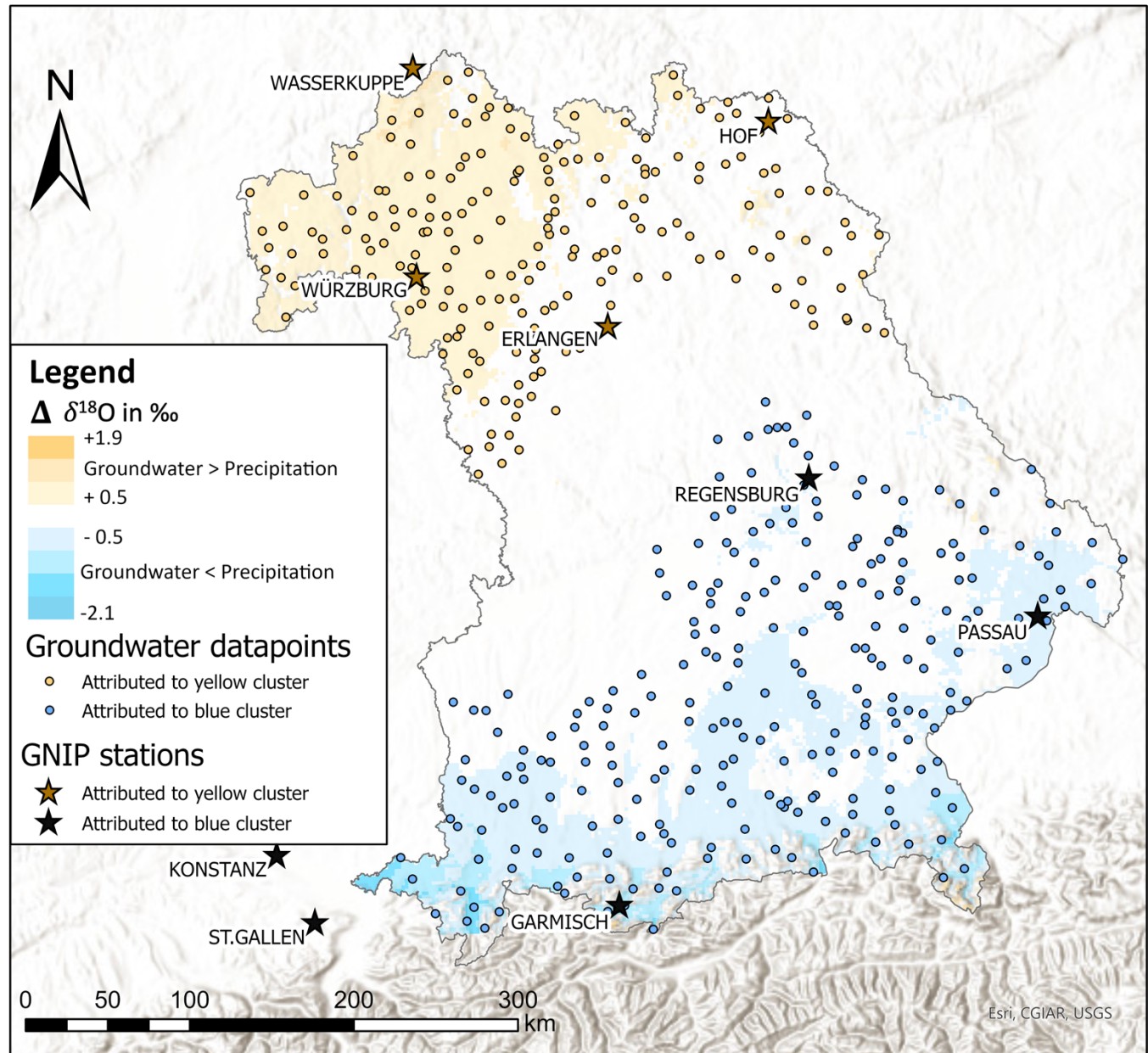

Figure 2: Differences between $\delta^{18}O$ values groundwater and precipitation isoscapes ($\Delta\delta$). Areas with differences smaller than |0.5 ‰| are transparent. The yellow and blue areas mark the clusters with groundwater–precipitation differences >|0.5 ‰|. The groundwater sampling points and GNIP stations attributed to one cluster according to Sect. 2.5 are also shown.

### 3.4 Regional Water Lines and d-values

Figure 3 shows the Regional Water Lines (RWL) for (a) the groundwater and precipitation data in the southern (blue) cluster and (b) groundwater and precipitation data in the northern (yellow) cluster.

In the southern region (Fig. 3a) the groundwater samples ($n$=262, with mean values of -10.1 ‰ $\delta^{18}$O and -73.0 ‰ $\delta^2$H) plot along a trendline with the following equation:

$$\delta^2 H_{\,gw,Alps} = (6.47 \pm 0.1)\, \delta^{18}O - (7.75 \pm 1.06) \tag{3}$$

The Southern Meteoric Water Line of the monthly precipitation values between 2010 and 2018 in the corresponding GNIP stations ($n$=420):

$$\delta^2 H_{\,prec,Alps} = (7.87 \pm 0.06)\, \delta^{18}O + (6.47 \pm 0.6) \tag{4}$$

In the northern region of Bavaria (Fig. 3b) the groundwater samples ($n$=202, with mean values of -9 ‰ $\delta^{18}$O and -63.9 ‰ $\delta^2$H)
plot along a trendline with the following equation:

$$\delta^2 H_{\,gw,Main} = (6.1 \pm 0.17)\, \delta^{18}O - (8.99 \pm 1.53) \tag{5}$$

The monthly precipitation values between 2010 and 2018 in the corresponding GNIP stations ($n$=432) are:

$$\delta^2 H_{\,prec,Main} = (7.78 \pm 0.07)\, \delta^{18}O + (5.95 \pm 0.65) \tag{6}$$

The reduced spread of groundwater isotope values in comparison to the corresponding precipitation values is most likely the result of subsurface mixing processes along the flow paths and with different water sources from different recharge areas and seasons.

When comparing the groundwater samples in the southern and northern clusters, we observe a clear trend towards less negative values going from south to north with minimum, maximum and mean $\delta^{18}$O values of -12.9, -6.8, and -10.1 ‰ and -10.6, -7.7, and -9.0 ‰, respectively. This observation reflects the same south-north trend as in the groundwater isoscape (Fig. 1).

The $d$ values in groundwater displayed in Fig. 4 range from 6.0 to 10.3 ‰ with higher values preferentially found at higher altitudes, although not consistently.

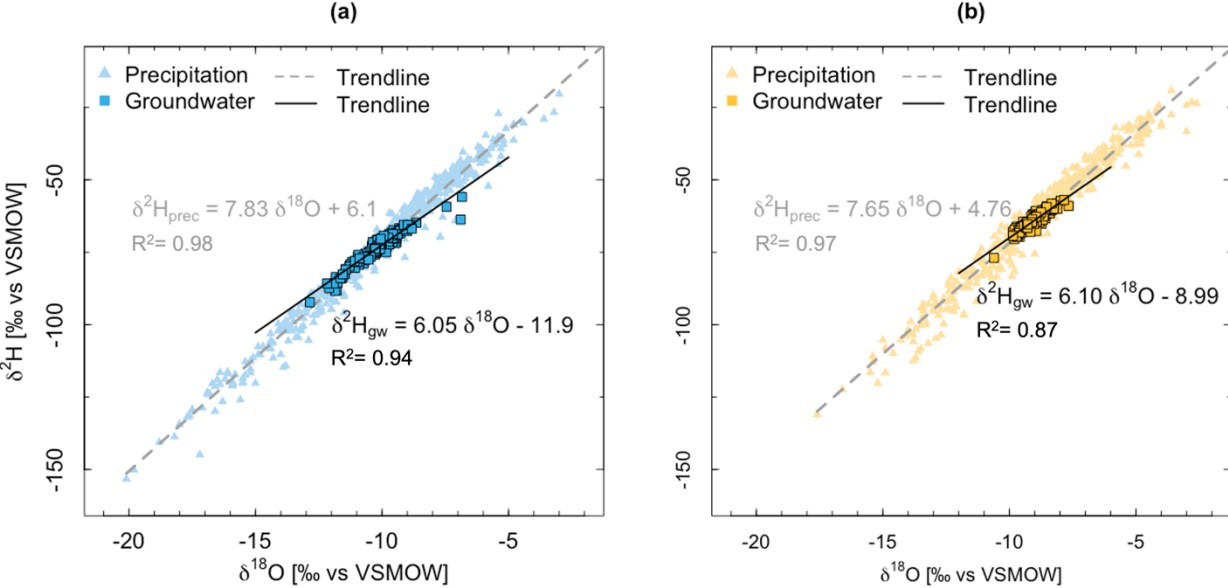


Figure 3: Regional Meteoric Water Lines (triangles) and Groundwater Lines (squares) (a) in the southern alpine and (b) northern Main regions.

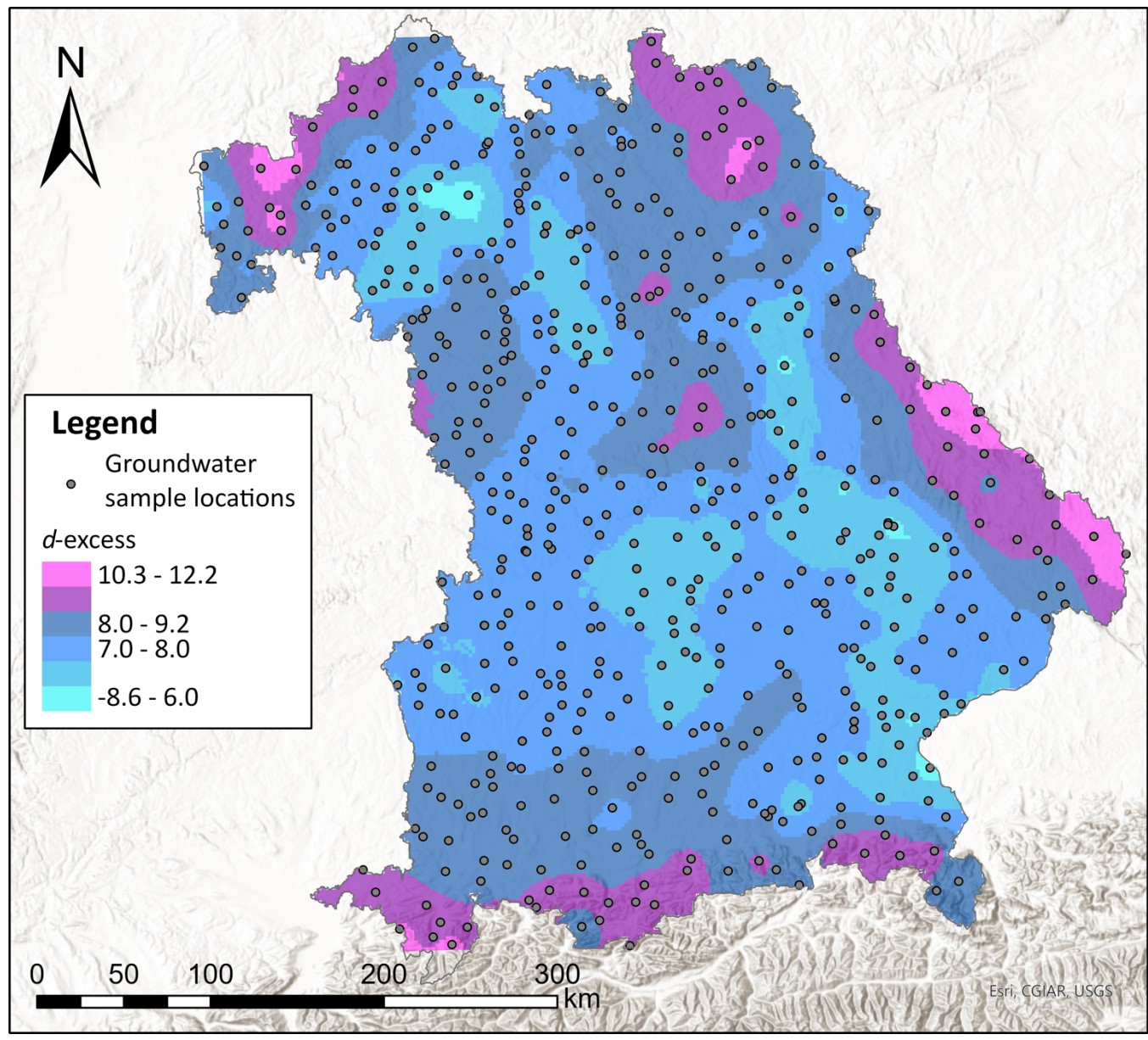

**Legend**

- Groundwater sample locations

*d*-excess

- 10.3 - 12.2
- 8.0 - 9.2
- 7.0 - 8.0
- -8.6 - 6.0

Figure 4: Interpolated *d* values in Bavarian groundwaters (obtained with empirical Bayesian Kriging) with sample locations.

## 4 Discussion

### 4.1 Isoscape comparison

With an average of 0.0084 wells/km², the presented groundwater dataset shows an exceptionally high density of points. For comparison, the isoscape produced by Regan et al. (2017) for Ireland is based upon 0.0025 wells/km² (factor 4 less) and the

data density of large-scale groundwater isoscapes for Mexico or Ethiopia lies one order of magnitude below (Wassenaar et al., 2009; Bedaso and Wu, 2021). Another advantage of our groundwater dataset is the even distribution of the sampling points across the study area that covers all major aquifer types in Bavaria, climatic regions, and altitudes up to 953 m a.s.l. It has to be noted that all groundwater sampling locations lie strictly within the boundaries of Bavaria and the interpolation shows higher prediction errors at the edges of the study site due to a lack of information outside of Bavaria. This effect is especially visible at the southern fringe of the study area, caused by larger differences among geographically close locations due to the larger heterogeneity of the alpine topography. The disadvantage of strict boundary cutoff is not present with the precipitation isoscape because it is retrieved from a Europe-wide interpolation.

The groundwater was sampled only once, within a short time frame (July 2015). The different groundwater values are therefore comparable. The isotope ratios in groundwater are expected to be representative for the annual average of isotopes in precipitation because neighbouring wells showed similar values across all regions. Additionally, the attenuation of seasonality is expected to take place in the unsaturated zone due to long enough travel times and even further dampened when infiltrating water is mixed with a much larger reservoir of existing groundwater with a wide range of ages. However, we cannot confirm that all sampled stations are representative for the annual averages of isotopes in precipitation. Therefore, we recommend investigating the seasonality of the isotope signal in shallow groundwater aquifers with quarterly sampling campaigns. Nevertheless, the observed trends remain robust because of the high number of consistent results across all regions and the generally low prediction error of the interpolation (< 0.3 ‰ $\delta^{18}O$). Thus, we can compare the produced groundwater isoscape to the precipitation map proposed by (Terzer et al., 2021). However, if there was a long-time trend in the isotope ratios in the precipitation, repeated groundwater measurements would be needed to assess such a trend.

## 4.2 Alpine region

The pattern observed in the southern region of Bavaria displays consistently more depleted $\delta^{18}O$ values in groundwater than in precipitation. This difference can be caused by predominant groundwater recharge during the colder season. Higher groundwater recharge rates in winter are expected for temperate climates, regions where snow melt is the dominating recharge process and because of little impact of evapotranspiration on the soil water balance in winter (Jasechko et al., 2014; Clark, 1997; Jasechko et al., 2017). Several studies in Europe have discussed a bias in groundwater isotopes towards winter recharge on local and regional levels (Raidla et al., 2016; Regan et al., 2017; Darling et al., 2003). Additionally, altitude effects are easily recognisable in alpine regions. When groundwater recharge areas lie at higher altitudes than the valley aquifers, groundwater is recharged with $\delta^{18}O$-depleted water. On a global average, altitude effects have been reported to affect the $\delta^{18}O$ value in precipitation by -0.2 ‰ per 100 meters elevation increase worldwide (Clark, 1997; Hemmerle et al., 2021) but even higher rates of -0.47 ‰ per 100 m have been reported in Germany (Stumpp et al., 2014). In our study area we calculated an altitude effect of -0.31 ‰ per 100 m in groundwater (supplementary information, Fig. S3). Moreover, a large part of the precipitation at higher altitudes falls as snow which further reduces its $\delta^{18}O$ values (Clark, 1997).

In the alpine regions it is therefore plausible that a combination of cold season groundwater recharge, snow and glacier melt dominated recharge, altitude and temperature effects explain the differences of up to 2.1 ‰ between the $\delta^{18}O$ values in precipitation and groundwater. The differences we report can be compared to other studies in mountainous areas. For example Cervi et al. (2017) found no differences between the means and medians in groundwater and precipitation in the Italian Alps.

On the other hand, a high altitude Swiss catchment showed more than 4 ‰ more negative groundwaters than local precipitation (Arnoux et al., 2020).

### 4.3 Main River region

In the northwest part of Bavaria, the Main River region, the groundwater displayed more positive $\delta^{18}O$ values when compared to precipitation. One possible explanation for this situation could be evaporation that occurs after precipitation reached the

ground. Such evaporation effects can strongly influence the groundwater isotopic composition. For instance, in British Columbia, Wassenaar et al. (2011) reported up to 4.5 ‰ $\delta^{18}O$ more positive groundwaters than precipitation. Higher evaporation rates are expected to occur in the Main region compared to the Alps due to the drier and warmer climate in these plains. The contribution of winter precipitation to groundwater is also less important in the Main region than in the rest of the study area both in terms of proportion and isotope signature. The yearly distribution of precipitation and the absence of a

significant snow cover as well as the less distinctive summer and winter isotope signals together with shorter winters all lead to a reduced winter bias in the groundwater isotopes.

Moreover, a factor that supports higher evaporation in the Main Region is the presence of widespread cropland in need for irrigation (43 % of the area in the Main region opposed to 35 % in the alpine region according to the land use in European Space Agency (2020)). This implies potentially large amounts of groundwater that is spread on the surface and becomes subject

to evaporation. Cherry et al. (2020) correlated the density of irrigation wells to the groundwater isotopic signal in Nebraska. They concluded that groundwaters in intensively irrigated areas showed an isotopic signature shift towards the non-growing season, whereas the rest of the state was not affected by this trend. In this study area, the origin and quantity of the water used for irrigation could not be quantified, yet we can usually consider irrigation is conducted with groundwater, therefore amplifying evaporation effects seen in groundwater.

Another factor that could plead for higher evaporation is the presence of the surface water body of the Main River which in turn partly feeds the groundwater. However, a 2013 study of the Main River shows that the $\delta^{18}O$ values vary between -10.0 ‰ in winter and -9.0 ‰ in summer and cannot explain the groundwater $\delta^{18}O$ values as far as 40 km from the river course (Türk, 2013). Moreover, no extensive artificial recharge is performed in the Main River region that would further increase the evaporation effect.

Despite these considerations (climate, irrigation and surface water feeding groundwater), the evaporation effect is barely seen in the RWL and $d$ values: the Regional Groundwater Line indeed displays a smaller slope than the precipitation line and indicates that evaporation might take place after precipitation reaches the ground. This slope difference is slightly larger in the Main region than in the Bavarian Alps, and further supports the hypothesis of a more pronounced evaporation effect in the

Main region. However, in our study case, the statistical differences between groundwater and precipitation do not allow to currently conclude on a significant evaporation effect. The $d$ values isoscape shows higher $d$ values with higher elevations. Nonetheless, no spatial correlations to the southern alpine or northern Main region could be observed. The distribution of $d$ values does not allow to conclude on different evaporative conditions between the Main region and the rest of the study area. This suggests that evaporation alone is not likely to be the main explanation for the 2 ‰ more positive $\delta^{18}O$ signal in the groundwater compared to precipitation in the Main region. A bias of the groundwater isotope signal towards the warm season precipitation due to summer groundwater recharge is not expected in our study area because the main groundwater recharge season in temperate central Europe remains winter.

Contributions from other aquifers (presumably deeper aquifers) may be a possibility if they add isotopically more positive waters. However, studies around Nuremberg by van Geldern et al. (2014) found more negative isotope values in deeper aquifers. A connection of the upper aquifer to deeper groundwater is unlikely in the Main region, as no deep aquifer is found under Northwest Bavaria (Büttner, 2003). Moreover, contributions via hydraulic windows would rather cause highly localized effects and do hardly explain the spatial extend of the effects found here. Similarly, we would not expect the influence of leeching tap water to influence the groundwater on the regional scale. Preferential flowpaths are another plausible mechanism to explain the observed pattern (Chen et al., 2023; Chen et al., 2024). They would apply to the scale observed in this study if heavy rain events with a more positive isotope signal (e.g. summer precipitation) influenced the aquifer. This is theoretically possible in the typical aquifers found in this region.

One further explanation for the isotopically heavier groundwater in the Main region can be attributed to the sampling time during the month of July 2015. Overall, stable isotopes in groundwater are assumed to show only little seasonal variations. Nonetheless, a lysimeter study by Stumpp et al. (2012) performed with Bavarian soils and climate reported that water isotopes show seasonality throughout the unsaturated zone; at least until a depth of 2 m below ground surface. On larger scales, Regan et al. (2017) also observed seasonal variation in groundwater at local scales in regions with rapid groundwater recharge. In the Main region, it is possible that the isotope signal in precipitation produces seasonal patterns in the groundwater because our sampling focused on shallow aquifers and the region is characterized by karst and fracture aquifers that provide favourable conditions for rapid transit times and low dispersivities. This might be an important and a so far hardly acknowledged reason for the up to 2 ‰ more positive groundwater $\delta^{18}O$ values when compared to precipitation. To investigate this aspect of groundwater isotopes, longer timeseries and at least quarterly sampling campaigns of both groundwater, surface water and precipitation would be necessary. So far, our data indicate scarce but potentially critical seasonality of large-scale groundwater recharge processes.

## 5. Conclusion and Outlook

In this study, we produced a high-resolution regional groundwater isoscape of $\delta^{18}O$ values in southern Germany. For the first time and as requested by previous studies, the data points were densely and evenly distributed across the study area and aquifer

types and all samples were analysed along the same strict protocols to reduce methodical uncertainties. The observed spatial pattern shows $\delta^{18}$O-depleted values in the southern alpine region and $\delta^{18}$O-enriched values in the Main region in the northwest of Bavaria. Further, comparisons between groundwater and precipitation isoscapes showed local and regional differences between these connected hydrologic compartments.

Differences in the southern Alpine region can be explained by the winter groundwater recharge, altitude effects and important snow cover. On the other hand, the more positive groundwaters in the northern Main River are more difficult to explain and hypotheses for this situation include evaporation after precipitation reached the ground, reduced winter groundwater recharge, possible connection to a deep-water aquifer and, perhaps most importantly, a bias due to the sampling season of the groundwater samples. Especially this last hypothesis would benefit from quarterly groundwater sampling campaigns in order

to investigate seasonality of stable isotopes in the upper aquifer on a regional scale. In addition, better documentation of the local irrigation practices and surface waters via stable isotopes would provide insights into the regional water balance.

This study shows that consideration of stable isotope values of both precipitation and groundwater are promising to outline and understand regional recharge mechanisms. Our results highlight the need for a better understanding of the seasonality of large-scale groundwater recharge patterns, based on quarterly or monthly sampling campaigns of all compartments of the hydrologic

cycle. Such investigations are gaining more importance with climate change. In Bavaria, the expected changes in the water cycle also due to climate change tend towards increased precipitation. However, with intense rain events in summer their uneven distribution throughout the year is expected to not contribute much to the groundwater recharge, a trend that has already been observed since several years already (Bayerisches Landesamt für Umwelt, 2022). Groundwater scarcity, especially in summer when needs are highest leads to increasing conflicts about its use. In this context, knowledge of the isotope composition

of groundwater helps to set this waterbody in relation to precipitation and surface waters. Our study proposes for the first time a distribution of background values for isotopes in groundwater which will also help frame local investigations about recharge quantification, groundwater isotopic tracing or use scenarios and support policymaking, resource management and environmental conservation efforts. Furthermore, this study may be replicated or widened to cover other and/or larger regions.

**Code and Data availability**

Original isotope data is available up on request.

**Competing Interests**

The authors declare that they have no conflict of interest.

**Author contribution**

Aixala Gaillard prepared the manuscript with contributions from all co-authors. Robert van Geldern and Johannes A. C. Barth
supervised the groundwater sampling and laboratory analyses. Christine Stumpp performed and provided the precipitation data. Aixala Gaillard applied the model and statistics.

**Acknowledgment**

This research was funded within the funding measure "Sustainable Groundwater Management" (LURCH) by the German Bundesministerium für Bildung und Forschung (BMBF, Project 'IsoGW', FKZ 02WGW1671A). We thank the Bayerisches
Landesamt für Umwelt and 17 Bavarian water management authorities for their participation as well as Tobias Juhlke for the logistic and organization of the sampling.

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
