# Peer review of "A new high-resolution groundwater isoscape for South-East Germany: insights from differences to precipitation"

_EGUsphere, 2024_

## Author Response (AR1)

**Response to reviewer, community and editor comments**

Journal: HESS

Title: " A new high-resolution groundwater isoscape for South-East Germany: insights from differences to precipitation "

Authors: Aixala Gaillard, Robert van Geldern, Johannes Arthur Christopher Barth, Christine Stumpp

Reviewer and community comments in *italic*; our answers in roman style.
Changes in the manuscript are highlighted by the MS Word tracking tool.

**Referee comments 1:**

*The study presents a high-resolution groundwater isoscape for southern Germany. This provides valuable insights into the mechanisms of groundwater recharge, particularly highlighting the differences in isotopic composition between alpine regions and lowland areas, which is crucial for water resource management. By comparing groundwater and precipitation isoscapes, the study offers a baseline for understanding how climate change may affect recharge patterns and, consequently, water availability. However, the groundwater samples were collected only once, during a short time frame. This limits the ability to assess seasonal variations in groundwater isotopes and long-term trends. The analysis of the reasons for the current distribution of groundwater isoscape and its significance for water resource management and utilization is not sufficiently discussed. In general, the paper requires a major revision to address the limitations and improve the interpretation of the results.*

*Major comments:*

*1. While the paper discusses evaporation as a potential explanation for the positive $\delta^{18}O$ values in the Main River region, it does not explore other possible mechanisms, such as the influence of aquifer types or preferential flowpaths.*

Contributions from other aquifers (presumably deeper aquifers) may be a possibility if they would add isotopically more positive waters. However, in studies around Nuremberg (Van Geldern et al., 2014) we found more negative isotope values in deeper aquifers. Moreover, contributions via hydraulic windows would rather cause highly localized effects and do not explain the larger extent of the effects found here. Preferential flowpaths of for instance rapid summer precipitation infiltration are another plausible mechanism. They would apply the scale observed in this study if heavy rain events with a more positive isotope signal (e.g. summer precipitation) influenced the aquifer. This would be possible in the typical aquifers found in this region.

The above are valid points that we have taken up in the discussion.

*2. It would be better if the author could add the quantification of the sources of groundwater. The spatial differences in the amount of precipitation infiltration to recharge the groundwater can help explain the study results.*

We have added numbers for the groundwater recharge to the manuscript based on the groundwater recharge map for the years 1981-2010 provided by the environmental office of Bavaria in Bayerisches Landesamt Für Umwelt (2018) and Bayerisches Landesamt Für Umwelt (2020).

*3. The manuscript would benefit from a more detailed discussion on the implications of the findings for water management strategies under changing climate conditions.*

In Bavaria, the expected changes in the water cycle also due to climate change point to generally more precipitation. However, with intense rain events in summer their uneven distribution throughout the year is not expected to contribute much to the groundwater recharge, a trend that is observed since several years already (Bayerisches Landesamt Für Umwelt, 2022). Groundwater scarcity, especially in summer when needs are highest leads to increasing conflicts about its use. In this context, knowledge of the isotope composition of groundwater helps to set this waterbody in relation to precipitation and surface water. Our study proposes for the first time a distribution of background values for isotopes in groundwater which will also help frame local investigations about recharge quantification, groundwater isotopic tracing or use scenarios. Such aspects support policymaking, resource management and environmental conservation efforts. Furthermore, this study is planned also on a nationwide scale in order to cover other regions.

These points have been taken up in the conclusion.

*4. Add a concept figure to summarize the impact of multiple factors such as altitude, irrigation conditions, snowmelt supply, and multi-source interactions on the groundwater isoscape.*

A concept figure of the discussed factors for both the alpine and Main River region has been added as graphical abstract.

***Specific comments:***

*L38, "Parallel to this global approach, several other local and regional interpolations of isotopes in precipitation have been proposed": What's the interpolation method?*

The most frequently used interpolation methods are regionally developed models that best fit the local parameters.

*L91: "70,500", "2,962".*

Changed.

*L106, "While few deep wells were also sampled": which ones? Add more details about the groundwater samples.*

We apologize for the misunderstanding. The deep wells discussed here still only represent the upper aquifer. Only scarce information in the well's filter depths is available, therefore we assumed a well-mixed water body in this study. Some clarifications were added to the text.

*L111: revise the orders for the following equations.*

We verified the numbering of the equations and found them to be correct,

*L174, "Meteoric water lines": The term 'Meteoric water lines' mainly refers to the linear relationship for precipitation isotopes. Is the expression 'Meteoric water lines for the groundwater' appropriate?*

We renamed the meteoric waterlines pertaining to groundwater as "groundwater lines"

*L175: meteoric waterlines, meteoric water lines, meteoric water line: please use the same terminology.*

Changed.

*L180, "This is similar to the MWL…": The differences of slope and intercept are obvious.*

The reviewer is correct, we removed the comparison.

*L184, "The reduced spread of groundwater isotope values in comparison to the corresponding precipitation values is most likely…": what is the main of this reduced spread?*

We assume the reviewer meant the "mean" of the groundwater data, which is -9.0 ‰ $\delta^{18}$O, -73.0 ‰ $\delta^2$H and -10.1 ‰ $\delta^{18}$O, -63.9 ‰ $\delta^2$H for the alpine and Main River regions respectively. This information was added to the Results section.

*L196: d-values, d-excess, please use the same terminology.*

Changed.

*L226, "On a global average, altitude effects have been reported to affect the δ18O": Is this altitude effect found for precipitation or groundwater?*

In the first instance it applies to precipitation and usually these effects are transferred to local groundwaters. The numerical values provided here refer to precipitation, which has been added to the manuscript.

*L232, "These differences range between findings from…": This is a wrong conclusion. Here, the groundwater isotope is more negative than the precipitation (up to 2.1‰), while the following is that the groundwater enriched to about 4.5‰.*

*L232-234: revise the sentence, pay attention on the uses of parentheses.*

Both comments on L232-234 have been addressed by revising the paragraph and making clear that here we present several examples for comparison to confirm similar trends. The reviewer is correct in highlighting that one of the provided references here is incorrect. It has been replaced.

*There are several papers focusing on exploring the isotope hydrology, please cite. Especially regarding how to use isotope information to quantitatively distinguish water sources and the impact of aquifer heterogeneity and preferential flow processes on isotope distribution and water flow, this is also a part that can be further discussed and analyzed in this study.*

*Chen, X., Yu, Z., Yi, P., Aldahan, A., Hwang, H.-T., and Sudicky, E.A. 2023. Disentangling runoff generation mechanisms: combining isotope tracing with integrated surface/subsurface simulation. Journal of Hydrology, 617, 129149.*

*Chen, X., Yu, Z., Yi, P., Hwang, H.-T., Sudicky, E.A., Tang, T., and Aldahan, A. 2024. Effects of soil heterogeneity and preferential flow on the water flow and isotope transport in an experimental hillslope. Science of the Total Environment, 917, 170548.*

We added both citations in the discussion.

*Figure 2: Groundwater and GNIP symbols are easily confused. It is recommended to replace the dots of GNIP with stars or other easily distinguishable symbols.*

Changed.

*References: L21, L34 "Jason B. West, 2008", L68 "H. S. Wheater, 2010", ..., L355, ...: Unify reference format.*

Changed.

**Referee comments 2:**

*The paper presents interesting results with broad applicability across various fields. Specifically, understanding the isotopic composition of groundwater can significantly enhance isotope-enabled hydrological models. This knowledge can also aid in the development of detailed maps, providing valuable insights into regional water cycles. Furthermore, the study serves as a robust case study that can be replicated in other regions, offering a template for similar research worldwide.*

*As a general comment, by incorporating these findings into the discussion or conclusion sections, the paper can highlight the practical implications and potential applications of the research. This addition would underscore the study's contribution to advancing hydrological science and its relevance to real-world environmental challenges. It would be also good to emphasize the usefulness of the research, showcasing its potential to inform policy-making, resource management, and environmental conservation efforts.*

All mentioned topics have been taken up into the discussion and conclusions to support the potential applications of the study.

***Some minor comments:***

*Paragraph 60: Infiltration of tap water or other artificial recharge (mentioned as for irrigation purposes) can explain the higher isotopic values in groundwater.*

We added the reviewer's considerations about tap water into our introduction and discussion although we would expect it to influence groundwater isotopic composition only locally and not throughout the entire Main Region. No large-scale artificial recharge is performed in the region which we now also specified in the text.

*Geological map will be useful to add to the study area description together with the groundwater table depth.*

As the reviewer suggests, a map displaying the main hydrogeological units in Bavaria was added in Figure 1 and referred to in the study site section.

*Paragraph 175: "regional meteoric water lines" – the terminology should be similar (later, it is written with capital letters).*

Changed, see comment on L174 by reviewer 1.

*Paragraph 210: please add several references on groundwater isotopic seasonality (e.g in Europe).*

To our knowledge, no studies have investigated groundwater isotope seasonality in Europe in regions that were not influenced by surface waters . Some studies have quantified stable isotope variations in soils, which have been cited in the article (e.g. Stumpp et al., 2012).

*Paragraph 245: Can higher isotopic values be attributed to the karst system and/or wetlands? Additionally, could the hydrogeology that facilitates summer rain recharge play a role? I noticed this discussion point appears later in the text (at the end of the discussion section) and suggest moving it up for a clearer understanding. Including references would also be beneficial.*

As the reviewer suggests, land-use, geology and hydrogeology may facilitate a quick response of the groundwater to rain events. Such forms of groundwater recharge have been addressed earlier in the text and some references were added.

*Can glaciers melt recharge groundwater in glacierized catchments?*

We included considerations of glacier melts in the snow melt effect on isotope composition in groundwater.

*The raw database can be useful to add.*

The data will be available through a nationwide ongoing research project for groundwater isotope data in Germany.

**Community comments 1:**

*General comments*

*Good research on groundwater hydrology that needs minor details before publication. Please, follow my specific comments to easily fix the issues.*

*Specific comments*

*Lines 21-22. "For groundwater studies, they can provide new information about the origin of precipitation or about surface water-groundwater interactions". Insert recent literature on H/O isotopes in groundwater:*

*- Lorenzi, V., Banzato, F., Barberio, M.D., Goeppert, N., Goldscheider, N., Gori, F., Lacchini, A., Manetta, M., Medici, G., Rusi, S. and Petitta, M., 2024. Tracking flowpaths in a complex karst system through tracer test and hydrogeochemical monitoring: Implications for groundwater protection (Gran Sasso, Italy). Heliyon, 10(2).*

*- Ju, Q., Hu, Y., Liu, Q., Chen, K., Zhang, H. and Wu, Y., 2024. Multiple stable isotopes and geochemical approaches to elucidate groundwater reactive transport paths in mining cities: A case from the northern Anhui, China. Science of The Total Environment, 912, 169706.*

The proposed references have been added to the relevant topics.

*Line 95. "the Molasse Basin covers combination of fracture and pore aquifers". Better something like "The Molasse Basin is characterized by aquifer units with flow occurring though either fractures or matrix"?*

Changed.

*Lines 122-132. Insert the Km2 for the study area. That's to remark that yours is a large-scale study.*

We indicated the area of our study site (70,500 km$^2$) in chapter 2.1 "Study site" only 3 paragraphs above the proposed mention and therefore believe it would be redundant to mention it again when addressing the interpolation methods.

*Lines 104-155. Details on sample preservation are missing.*

Samples were generally cooled during their transport and storing. No additives or preservation methods have been used. However, longtime experiences in our laboratories have shown that stable isotope ratios are not affected by different conditions as long as they are kept tight. This is already implied in the Methods section.

*Line 126. "Empirical Bayesian Kriging". You need to link the method of interpolation to figures 1 and 4.*

A reference to the method has been added in the figure captions.

*Line 289. "Changing weather". Better "Climate change"?*

Changed.

*Line 305. Update the reference list with recent literature on the topic.*

See response to reviewer 2, paragraph 210.

*Figures and tables*

*Figure 1. "A" and "B should be re-versed because the figure of Europe is the one at larger scale with no data plotted.*

Changed.

*Figure 1. Specify that the Empirical Bayesian Kriging was used. Insert the information in the caption.*

Changed, see response to comment on line 126.

*Figure 3. Do you need to provide R2 for the four lines?*

The correlation coefficients range between 0.87 and 0.98 for the four lines, indicating excellent linear correlation. This information was added in the figure.

*Figure 4. Specify that the Empirical Bayesian Kriging was used. Insert the information in the caption.*

Changed, see response to comment in line 126.

**Editor comments:**

*Two reviewers provided constructive comments on the paper. Authors addressed these in responses, and are encouraged to further elaborate the new contribution/finding of this study, and discuss the uncertainty associated with the data and analysis in the revision.*

We added precisions about the interpolation uncertainty in the Methods (section 2.4), extended the comparison on the error associated to the data vs the observed trends and added considerations about the uncertainty of the interpolation in the Results (section 3.1) and further explained observations on the interpolation uncertainty in the discussion (section 4.1).
Moreover, we highlighted the novelty this study provides with regard to data density and the hydrological interpretations such isoscapes allow.

**Futher Added Bibliography**

Bayerisches Landesamt für Umwelt: Mittlere Jährliche Grundwasserneubildung in Bayern 1981-2010, Bayerisches Landesamt für Umwelt (LfU), Augsburg, 2018.

Bayerisches Landesamt für Umwelt: Mittlere Grundwasserneubildung in den Bezirken der bayerischen Wasserwirtschaftsämter, Bayerisches Landesamt für Umwelt, Augsburg, 1-2, 2020.

Bayerisches Landesamt für Umwelt: Sicherheit der Wasserversorgung in Not-,Krisen- und Katastrophenfälle, Bayerisches Landesamt für Umwelt,, Augsburg, 2022.

Stumpp, C., Stichler, W., Kandolf, M., and Šimůnek, J.: Effects of Land Cover and Fertilization Method on Water Flow and Solute Transport in Five Lysimeters: A Long-Term Study Using Stable Water Isotopes, Vadose Zone Journal, 11, https://doi.org/10.2136/vzj2011.0075, 2012.

van Geldern, R., Baier, A., Subert, H. L., Kowol, S., Balk, L., and Barth, J. A. C.: Pleistocene paleo-groundwater as a pristine fresh water resource in southern Germany – evidence from stable and radiogenic isotopes, Sci. Total. Env., 496, 107-115, https://doi.org/10.1016/j.scitotenv.2014.07.011, 2014.

---

## Author Response (AR2)

**Response to reviewer and editor comments**

Journal: HESS

Title: " A new high-resolution groundwater isoscape for South-East Germany: insights from differences to precipitation "

Authors: Aixala Gaillard, Robert van Geldern, Johannes A. C. Barth, Christine Stumpp

Reviewer and community comments in *italic*; our answers in roman style.
Changes in the manuscript are highlighted by the MS Word tracking tool.

**Referee comments 1:**

*The authors have made sufficient modifications to the previous round of revision. The groundwater isoscape proposed in the paper is very meaningful. However, the discussion of the reasons for isotope distribution is still relatively speculative, lacking some quantitative and targeted analysis. Here, before the paper can be accepted, I suggest the authors can make moderate revisions to further enhance the quality, clarity, and overall contribution of the paper.*

***Main comments:***

*1. In section 4.2, the author discussed the influence of altitude on isotope distribution and cited quantitative results from other studies. It is suggested that based on the altitude changes and isotope distribution in this study, the isotopic altitude effect in the study area can be quantitatively calculated. It is possible to consider adding a graph of isotopic changes with altitude and comparing it with other studies.*

Altitude effects on isotope distribution are well known and described in precipitation. This has been done in Germany based on the GNIP dataset by Stumpp et al. (2014), which we cite. Therefore, we do not repeat the calculations in this study. For groundwater, the altitude effect amounts to -0.31‰ ($\delta^{18}O$) per 100 m elevation gain over the entire study area. A Figure with this correlation has been added to the supplementary materials and mentioned in section 4.2.

*2. In section 4.3, the author discussed multiple influencing factors of isotope distribution, and it is better to add more quantitative analysis. For example, the author mentioned the evaporation effects. It is necessary to add the specific spatiotemporal variations of evaporation in mountainous and plain areas of the study area to help better understand the evaporation effect.*

We rearranged section 4.3 according to the wishes of the reviewer, highlighting the climatic and topographic differences between the alpine region and the Main river regions in the first paragraph (lines 252-259) before discussing the factors specific to the Main River Region which is the focus of the section.

***Some specific comments:***

*Graphical abstract, "-10.4 δ18O" should add the unit of "‰".*

Changed

*L119: revise the orders for the following equations. The equation 1 is in the L82.*

Changed

*L352: revise the sentence: "We thank the We thank the...".*

Changed

*L367: revise the format of references: "Bayerisches Landesamt für Umwelt,, Augsburg,".*

Changed

*Further check for other writing and formatting issues throughout the entire text.*

We revised mainly unit and citation formats, numbering and δ-notations.

**Referee comments 2:**

*The manuscript „A new high-resolution groundwater isoscape for South-East Germany: insights from differences to precipitation" by Gaillard et al. presents for the first time a regionalized groundwater isotope map based on 596 measurements for south-east Germany. The work not only reflects on groundwater isotope patterns, but also compares to local precipitation patters. This work, including the applied techniques and the attributed data set will definitively attract future attempts to sustainably judge and manage groundwater resources. I agree with the prior comments to the work and think that the authors have commented and improved almost all issues.*

*I recommend acceptance after only minor revision.*

**Minor comments:**

*Line 83: (Earlier comment by referee #1): Line 111: revise the orders for the following equations. Please check again, because you use „equation (1)" in the introduction chapter (Line 83, revised version) and again „equation (1)" (Line 119) in the Methods chapter. You need to re-number to equations 2-6 and check references to equations in the text.*

Changed

*Lines 34, 130, 172, 237, 474: Terzer-Wassmuth et al. (for 2021, but Terzer et al. 2013 is correct!)*

Changed where necessary. When referring to the produced precipitation isoscapes that we used in this study, we only mention the latest update, after explicitly citing the underlying work of Terzer, 2013 on first occurrence.

**Editor comments:**

*Two reviewers provide positive comments on the paper. Authors are required to address all comments on the revision, and have a thorough editing on the revision.*

We addressed all revisions suggested by the reviewers.